# The Impact of Nutrition-Based Interventions on Nutritional Status and Metabolic Health in Small Island Developing States: A Systematic Review and Narrative Synthesis

**DOI:** 10.3390/nu14173529

**Published:** 2022-08-26

**Authors:** Eden Augustus, Emily Haynes, Cornelia Guell, Karyn Morrissey, Madhuvanti M. Murphy, Cassandra Halliday, Lili Jia, Viliamu Iese, Simon G. Anderson, Nigel Unwin

**Affiliations:** 1The Faculty of Medical Sciences, The University of the West Indies, Cave Hill Campus, P.O. Box 64, Bridgetown BB11000, Barbados; 2European Centre for Environment and Human Health, University of Exeter, Truro TR1 3HD, UK; 3Division of Sustainability, Society and Economics, Department of Technology, Management and Economics, Technical University of Denmark, Produktionstorvet 358, DK-2800 Kgs. Lyngby, Denmark; 4The George Alleyne Chronic Disease Research Centre, Caribbean Institute of Health Research, The University of the West Indies, Bridgetown BB11000, Barbados; 5Institute for Manufacturing, University of Cambridge, Cambridge CB3 0FS, UK; 6Pacific Centre for Environment and Sustainable Development, University of the South Pacific, Suva 0101, Fiji; 7Glasgow-Caribbean Centre for Development Research, University of the West Indies, Bridgetown BB11000, Barbados; 8MRC Epidemiology Unit, University of Cambridge, Cambridge CB2 0QQ, UK

**Keywords:** nutritional status, metabolic health, small island developing states

## Abstract

Small island developing states (SIDS) have a high burden of nutrition-related disease associated with nutrient-poor, energy-dense diets. In response to these issues, we assessed the effectiveness of nutrition-based interventions on nutritional status (under-nutrition) and metabolic health (over-nutrition) among persons in SIDS. We included SIDS-based nutrition studies with change in nutrition status (e.g., markers of anaemia) or metabolic status (e.g., markers of glycaemia) as outcomes. The PRISMA framework was applied and MEDLINE, Embase, CINAHL, OARE library, Web of Science, Scopus, ASSIA, EconLit, AGORA, AGRICOLA, AGRIS, WHO-EMRO, and LILACS were searched (2000–2020). Cochrane risk of bias (ROB) and Cochrane ROBINS-I tools assessed ROB for randomised and non-randomised studies, respectively. PROSPERO registration (CRD42021236396) was undertaken. We included 50 eligible interventions, involving 37,591 participants: 14 trials reported on nutritional status, 36 on metabolic health. Effective interventions, evaluated at the individual level, took a multifaceted approach for metabolic outcomes; while nutrition outcomes utilised supplements. Most intervention types were suitable for issues related to ‘over’ nutrition versus ‘under’ nutrition. Twenty-six studies (nutrition status (six); metabolic health (twenty)) were effective (*p* < 0.05). With the current rise of nutrition-related public health challenges, there is a need for further development and evaluation of these and related interventions at the population level.

## 1. Introduction

Small island developing states (SIDS), home to over 69 million people across 58 countries and territories, have an alarming legacy of a complex history of malnutrition, economic, and environmental vulnerabilities, with a looming health crisis [1,2,3]. The impact of these vulnerabilities coupled with urbanisation, internal migration, and the effects of international trade agreements on agriculture, has decreased local agricultural production and increased reliance on imported foods [1,4]. Food imports, which constitute ultra-processed, energy dense, and nutrient-poor products, have risen over the past decades from over a third to almost two thirds of what is consumed within SIDS in the Pacific and Caribbean [5,6]. 

These complex issues lead to food and nutrition insecurity and are associated with a shift from diets based on whole foods, such as local fruits, vegetables, roots, and tubers [6,7,8]. The downstream effect is a high burden of nutrition-related non-communicable diseases (NCDs), such as obesity, type 2 diabetes, cardiovascular diseases, micronutrient deficiencies, and persistent childhood stunting and wasting in many territories [9]. In the Caribbean, one third of the population has been categorised as obese while three quarters of adult mortality in the Pacific is attributed to NCDs, 76% of which is premature (adults 30 to 69 years of age) [5,10]. Stunting still exceeds 20% in five of the poorer SIDS, and wasting still strikes a concern within some countries (>10%) [5].

To address these public health concerns, in 2014, the United Nations (UN) Food and Agricultural Organization (FAO), in collaboration with relevant government bodies within the SIDS, developed a plan of action underscoring the need for policies and interventions to decrease nutrition-related disease and develop resilient, nutrition-sensitive food systems [11,12,13,14,15]. This led to the development of the Global Action Programme on Food Security and Nutrition for SIDS (GAP) in 2017 [12,16]. The GAP, which complements the SIDS Accelerated Modalities of Action (SAMOA) Pathway, emphasises the importance of an integrated approach to sustainable development, as highlighted in the Sustainable Development Goals (SDGs) 2030 Agenda [5,12,15,16]. Its framework relates to three objectives surrounding food security, involving the development of programs and interventions across entire food systems to promote capacity building aimed at empowering communities, and a participatory and sustainable approach aimed at building resilience within local food systems with the consideration of environmental challenges [5,15].

A recent systematic scoping review that examined the health-related impact of community food initiatives in SIDS, concluded that approaches utilised within these settings were inconsistent, owing to heterogeneity in study outcomes that hindered synthesis and applicability of the evidence. The review also highlighted that few studies included a description of any theoretical frameworks to hypothesise or explain the impacts of community food initiatives and that most research within these settings focused on the environmental impact of coastal and marine resources [17]. However, this review did not appraise study quality, or critically draw out the details of the study findings, and highlighted that there was sufficient literature to answer systematic review questions with the identified evidence gaps due to an absence of literature. Hence, we have tried to identify best practices and approaches by systematically reviewing the interventional evidence. This systematic review is the first to assess the impact of nutrition-based interventions aimed at improving nutritional status and metabolic health in SIDS. It aims to highlight evidence needed to guide interventions, programmes, and policies in SIDS settings. 

## 2. Methods

We systematically reviewed nutrition-based interventions aimed at improving metabolic health and nutrition status among persons in SIDS. Metabolic health outcomes are defined as those related to over-nutrition, such as obesity, type 2 diabetes, and heart disease, and nutritional outcomes are defined as indicators related to under-nutrition, including nutrient deficiencies, stunting, and wasting. This review protocol was registered with the International Prospective Register of Systematic Reviews (PROSPERO) (CRD42021236396) and can be accessed at the following link: https://www.crd.york.ac.uk/prospero/display_record.php?RecordID=236396 (accessed on 5 July 2022). The review was conducted and reported in accordance with Preferred Reporting Items for Systematic Reviews and Meta-Analysis (PRISMA) [18].

The specific objectives of this review were: (i) to identify and review published and grey literature on interventions, including a nutrition component aimed at improving nutritional status and metabolic health of persons living in SIDS; (ii) to evaluate the quality of these studies by assessing the risk of bias; and (iii) to provide narrative, and, as appropriate, statistical summaries of the findings including the effect sizes and their precision. 

### 2.1. Eligibility Criteria

All studies that met the inclusion criteria in Table 1 were eligible for inclusion. There were six criteria: study design, outcomes, study setting, publication status, language, participant characteristics and time. Selected studies were set within a 20-year time frame, to ensure that emerging evidence was relevant in informing future interventions, programs, and policies. These were dated from 1 January 2000 to 1 August 2020, when the review began. 

### 2.2. Search Strategy

The search strategy was developed and piloted with assistance of a medical librarian from the University of the West Indies in August 2020. We used thirteen databases including those related to health, environmental and social science, agricultural science, and cross-disciplinary databases. Health related databases included Medical Literature Analysis and Retrieval System Online (MEDLINE) (via PubMed), Cumulative Index to Nursing and Allied Health Literature (CINAHL), Excerpta Medica dataBASE (EMBASE) (via Ovid) and Cochrane Library databases. Environmental and social science related databases included: Online Access to Research in the Environment (OARE) library, Web of Science: Conference Proceedings Citation Index, Science Citation Index Expanded, and Social Science Citation Index, Scopus, Applied Social Sciences Index and Abstracts (ASSIA) via ProQuest, and EconLit. Agricultural science related databases included AGRICOLA (US National Agriculture Library), Access to Global Online Research in Agriculture (AGORA), and International System for Agricultural Science and Technology (AGRIS) (both hosted by FAO). Regional and cross disciplinary databases included Latin American and Caribbean Health Sciences Literature (LILACS) and Afrolib (see Box S1, Appendix A). Reference lists of included studies were checked to identify any other potentially relevant studies (i.e., backward citation searching) as well as reference lists of other identified reviews, particularly systematic reviews that were conducted on similar topics or within SIDS settings. 

As highlighted in Table 1, no language restrictions were applied, however, search terms were written and applied to databases in English, and all databases were English language based. Study authors were contacted if there was insufficient information to assess eligibility such as an intervention or study conducted across other non-SIDS settings to enquire about non-aggregated data. 

### 2.3. Study Selection

All selected studies were uploaded into an online bibliographic database: Rayyan reference manager [20]. Duplicates were excluded manually via the statistical software Stata 16 (StataCorp, College Station, TX, USA) and through the Rayyan manual de-duplication option. Titles and abstracts were screened in duplicate by five pairs of reviewers (E.A., S.G.A.; S.W., E.H.; C.G., C.H.; M.M.M., N.U.; L.J., K.M.), and those that met the inclusion criteria were considered eligible for full-text screen. Articles were included for full-text screen if there was insufficient information to exclude them. Full texts were obtained and screened in duplicate by the same pairs, any discrepancies within the pairs were resolved by a third reviewer (V.I.). 

### 2.4. Data Extraction

Eligible full texts were extracted in duplicate by four pairs (E.A., S.W.; E.H., C.H.; C.G., N.U.; L.J., K.M.). An online data extraction form was developed, tested, and modified via Research Electronic Data Capture (REDCap), a secure online data collection platform [21]. This was made available to all reviewers. The data extraction form included three sections: publication details, study details, and risk of bias. The publication details included the source, type, and title of record, along with author and journal details. Study details included the intervention setting, intervention name, study design, sample population demographics, sample size (intervention and control group where relevant), intervention details (rationale/theory, procedures/components, dose/frequency, mode of delivery, timeframe), description of conditions for intervention or control groups, methods, tools used for data capture or health assessment, outcome description and type including information on baseline and follow up points, and lessons learnt by study authors. Extracted study details were downloaded to Microsoft Excel and discrepancies or conflicts were resolved by the third reviewer. 

### 2.5. Risk of Bias

Risk of bias assessments were completed in duplicate by four pairs (E.A., S.W.; E.H., C.H.; C.G., N.U.; L.J., K.M.). The quality of all included studies was evaluated by assessing their risk of bias using the Cochrane risk of bias tool for randomised trials, and the Cochrane ROBINS-I tool for non-randomised studies [22,23]. The tools assessed risk across several domains relevant to the design of the studies, including randomisation, deviation from intended intervention, missing data, measurement of outcome, and selection of reported result. For each included study, risk of bias was assessed for the primary outcome measure, as relevant to this review. In most cases, this was a metabolic outcome such as markers of glycemia including haemoglobin A1C (HbA1c), fasting blood glucose (FBG), change in mean blood glucose, insulin sensitivity (*n* = 12), Body Mass Index (BMI) (*n* = 10), weight change (*n* = 8), change in BP (*n*= 4), triglycerides (TAGS) (*n* = 1), and endothelial function (*n* = 1), with nutrition outcomes measured being z-scores (*n* = 7), markers of anaemia including iron absorption and haemoglobin concentration (*n* = 4), stunting (*n* = 1), vitamin D deficiency (*n* = 1) and malnutrition (*n* = 1).

### 2.6. Results Synthesis

Due of the heterogeneity of studies, related to intervention type, study design, participants, outcomes and measures, a meta-analysis to estimate a pooled effect size was not possible. Therefore, a narrative or descriptive analysis was undertaken. However, we used stratified analysis to assess levels of evidence of effect for studies related to their outcome measure, type of intervention, intervention level, and intervention intensity. 

## 3. Results

### 3.1. Study Characteristics

#### 3.1.1. Overview

Fifty unique records involving 37,591 participants were eligible and included for synthesis (Figure 1). One study was part of a multi-country project, with data analysed and reported separately [24] Figure 2 highlights included studies by geographic location, intervention level and type. Table 2 stratifies the studies by country, design, outcome measure, and effectiveness. An assessment of risk of bias is also included in Table 2. 

#### 3.1.2. Location and Design

Studies were conducted in seventeen countries. Twenty-four studies were conducted in the Caribbean: nine in Haiti [25,26,27,28,29,30,31,32,33], three in Cuba [34,35,36], three in Dominican Republic [37,38,39], three in Jamaica [24,40,41], three in Trinidad and Tobago [42,43,44], two in Puerto Rico [45,46] and one in Barbados [47]. Eighteen studies were conducted in the Atlantic, Indian Ocean, Mediterranean and South China Sea (AIMS region): fifteen in Singapore [48,49,50,51,52,53,54,55,56,57,58,59,60,61,62], two in Mauritius [63,64] and one in Seychelles [65]. Eight studies were conducted in the Pacific region: two in Fiji [66,67], one in American Samoa [68], one in Tonga [69], one in French Polynesia [70], one in Papua New Guinea [71], one in Kiribati [72], and one in Samoa [73]. 

Of the fifty studies, twenty-eight used randomised designs: fifteen individually-randomised parallel group trials [40,41,43,44,45,46,48,49,50,51,53,55,57,64,65], eight cluster-randomised parallel group trials [25,29,30,31,33,61,68,72], and five individually-randomised cross over or other matched trials [24,46,52,58,64]. Twenty-two were of non-randomised study design: fifteen uncontrolled before and after studies [32,34,36,37,39,47,52,54,56,57,59,62,63,67,71], six controlled before and after studies [26,38,60,66,69,70] and one interrupted time series and repeated measures study [73]. 

**Table 2 nutrients-14-03529-t002:** Overview of included studies’ country location, study design, outcome measures and risk of bias by intervention type (*n* = 50).

Ref.	Region/Country	Study Design	Nutrition/Metabolic	Measured Outcome	Evidence for Effectiveness	Risk of Bias
Specific Food						
[42]	Trinidad and Tobago	Non-randomised controlled before/after study	Metabolic	Blood pressure	+	
[48]	Singapore	Individually-randomised parallel-group trial	Metabolic	Insulin sensitivity	−	
[65]	Seychelles	Individually-randomised cross over trial	Metabolic	TAGS	+/−	
[58]	Singapore	Individually-randomised cross over trial	Metabolic	Fasting blood glucose	+	
Supplements/Fortified Foods						
[25]	Haiti	Cluster-randomised parallel group trial	Nutrition	Haemoglobin concentration	+	
[50]	Singapore	Individually-randomised parallel-group trial	Nutrition	Vitamin D deficiency	+	
[51]	Singapore	Individually-randomised parallel-group trial	Metabolic	Endothelial function	+	
[41]	Jamaica	Individually-randomised parallel-group trial	Nutrition	Weight for age/height z scores	−	
[27]	Haiti	Individually-randomised cross over trial	Nutrition	Iron absorption	+/−	
[28]	Haiti	Individually-randomised parallel-group trial	Nutrition	Stunting	+	
[29]	Haiti	Cluster-randomised parallel group trial	Nutrition	Weight for age/height z scores	+/−	
[30]	Haiti	Cluster-randomised parallel group trial	Nutrition	Anaemia	+/−	
[37]	Dominican Republic	Non-randomised uncontrolled before and after study (pre/post-test study)	Nutrition	Weight for age/height z scores	−	
[31]	Haiti	Cluster-randomised parallel group trial	Nutrition	Anaemia	+	
[64]	Mauritius	Individually-randomised parallel-group trial	Metabolic	HbA1C	+/−	
Nutrition Education						
[49]	Singapore	Individually-randomised parallel-group trial	Metabolic	BMI	+	
[63]	Mauritius	Non-randomised uncontrolled before and after study (pre/post-test study)	Metabolic	BMI	−	
[70]	Singapore	Individually-randomised parallel-group trial	Metabolic	Weight change	−	
[54]	Singapore	Non-randomised uncontrolled before and after study (pre/post-test study)	Metabolic	Weight change	+	
[38]	Dominican Republic	Non-randomised controlled before/after study	Nutrition	Weight change for age/height z scores	+	
[72]	Kiribati	Cluster-randomised parallel group trial	Metabolic	Change in mean blood glucose	−	
Multifaceted Intervention						
[68]	American Samoa	Cluster-randomised parallel group trial	Metabolic	HbA1C	+	
[69]	Tonga	Non-randomised controlled before/after study	Metabolic	Weight change	+/−	
[70]	French Polynesia	Non-randomised controlled before/after study	Metabolic	Weight change	+/−	
[45]	Puerto Rico	Individually-randomised parallel-group trial	Metabolic	BMI	+	
[26]	Haiti	Non-randomised controlled before/after study	Nutrition	Weight change for age/height z scores	+/−	
[52]	Singapore	Non-randomised uncontrolled before and after study (pre/post test study)	Metabolic	HbA1C	+	
[66]	Fiji	Non-randomised controlled before/after study	Metabolic	BMI for age	−	
[71]	Papua New Guinea	Non-randomised uncontrolled before and after study (pre/post test study)	Nutrition	Weight change for age/height z scores	+	
[55]	Singapore	Individually-randomised parallel-group trial	Metabolic	Weight Change	−	
[44]	Trinidad and Tobago	Individually-randomised parallel-group trial	Metabolic	BMI	−	
[67]	Fiji	Non-randomised uncontrolled before and after study (pre/post test study)	Metabolic	Blood pressure	+	
[33]	Haiti	Cluster-randomised parallel group trial	Nutrition	Weight change for age/height z scores	+/−	
[57]	Singapore	Non-randomised uncontrolled before and after study (pre/post test study)	Metabolic	Weight change	+	
[73]	Samoa	Non-randomised interrupted time series studies and repeated measures study	Metabolic	Weight change	−	
[59]	Singapore	Non-randomised uncontrolled before and after study (pre/post test study)	Metabolic	BMI	+	
[60]	Singapore	Non-randomised controlled before/after study	Metabolic	BMI	+	
[35]	Cuba	Individually-randomised parallel-group trial	Metabolic	BMI	+/−	
[36]	Cuba	Non-randomised uncontrolled before and after study (pre/post test study)	Metabolic	Weight change	+/−	
[39]	Dominican Republic	Non-randomised uncontrolled before and after study (pre/post test study)	Metabolic	HbA1C	+	
[61]	Singapore	Cluster-randomised parallel group trial	Metabolic	Blood pressure	+	
[62]	Singapore	Non-randomised uncontrolled before and after study (pre/post test study)	Metabolic	HbA1C	+	
Policy						
[32]	Haiti	Non-randomised uncontrolled before and after study (pre/post test study)	Nutrition	Severe childhood malnutrition	−	
Dietary Change						
[43]	Trinidad and Tobago	Individually-randomised parallel-group trial	Metabolic	BMI	+	
[40]	Jamaica	Individually-randomised parallel-group trial	Metabolic	HbA1C	+	
[47]	Barbados	Non-randomised uncontrolled before and after study (pre/post test study)	Metabolic	Fasting blood glucose	+	
[24]	Jamaica and Nigeria	Individually-randomised cross over trial	Metabolic	Blood pressure	+	
[46]	Puerto Rico	Individually-randomised parallel-group trial	Metabolic	BMI	−	
[34]	Cuba	Non-randomised uncontrolled before and after study (pre/post test study)	Metabolic	Fasting blood glucose	+/−	
[56]	Singapore	Non-randomised uncontrolled before and after study (pre/post test study)	Metabolic	HbA1C	+	

+ mostly significantly effective on outcomes measured; +/− some significant positive effects plus some/no change/insignificant effect; − no significant positive effects or negative effect. Region: Caribbean, Pacific, AIMS. Overall ROB by study: low risk, moderate risk, high risk. Table is stratified by intervention type (specific food, supplements/fortified food, nutrition education, multifaceted, policy and dietary change).

#### 3.1.3. Intervention Types

Among the selected studies, several types of nutrition-based interventions were identified which assessed an aspect of nutritional status (‘under-nutrition’) (*n* = 14) or metabolic health (‘over-nutrition’) (*n* = 36) as an outcome measure. These intervention types were associated with variations related to intervention level, the group targeted or the setting. 

Most interventions were multifaceted (*n* = 21) including aspects of nutrition education, combining dietary change, supplements, and non-nutritional components including physical activity, behavioural therapy, and self-medication management (Appendix A). Eight were implemented at the individual level, targeting adults [35,36,57,62], children [26,71] or solely women [52,55] within a clinical setting. Two interventions were implemented at the household level, one targeted adults within a clinical setting [68] and one targeted children within a community setting [33]. Five were implemented at the institutional level, three targeted children within a school setting [44,45,70], and two targeted adults within the workplace [59,60]. Four were implemented at the community level, targeting children at school [69] and within the community [66], and adults within the community [61] or general population [39]; two were implemented at the national or policy level, targeting adults within the general population [67,73]. 

Eleven interventions focused on the use of supplements or fortified foods. Seven were implemented at the individual level and included interventions that targeted adults within the general population [50,64] or clinical setting [51], and children in a clinical or community setting [28,31]; two were implemented at an institutional level targeting children at school [29,30]; and two interventions were implemented at the community level that targeted children in a community setting [25,27]. Of the eleven, nine were supplement-based, including micronutrient sprinkles and powder to improve anaemia [25,31]; zinc [41], lipid [28] and food [29] supplementation to improve linear growth and body composition; and vitamin A and Folic Acid supplements to decrease protein energy malnutrition (PEM) among children [30]. Other supplement-based interventions included vitamin D [50], cholecalciferol [51], and natural antioxidant [64] supplements to improve nutritional status and oxidative stress among adults including the elderly and type 2 diabetics. Two interventions were based on fortified foods, including ferrous fumarate fortified wheat [27], and a fortified snack [30] to improve anaemia and body composition among children. 

Seven interventions included an aspect of dietary change. All were implemented at the individual level among adults targeted within a clinical [40,43,46,56] or general population/public [24,34,47] setting. Dietary changes included the use of a low-carbohydrate (ketogenic) diet [43], high calcium diet [46], and a vegetarian macrobiotic diet [34] to improve BMI; a glycaemic-index adapted Caribbean based diet [40], a liquid low-calorie diet [47], and fasting [56] to improve markers of glycemia; and a modified salt diet [24] to assess impact on blood pressure. 

Six interventions focused on nutrition education. Five were implemented at the individual level and included education that targeted adults [49,54], or solely women in a clinical setting [53]; housewives [63] or children in a clinical setting [38]; and one intervention was implemented at the institutional level targeting children in a school setting [72]. These six nutrition education interventions provided face to face sessions on food choices including calorie content, food groups and serving sizes to support healthy weight [49,63], pre- and post-natal [38] and sugar reduction lectures [72]; as well as ambulatory nutrition support via telephone [54], and food coaching from an app [53]. 

Four interventions focused on specific foods. All were implemented at the individual level and targeted adults within their home [48], the Ministry of Health [65], or general population [58]. One intervention had no setting information [42]. These interventions included using cocoa [42]), *n*-3 enriched polyunsaturated fatty acid (PUFA) eggs [65], coffee [48], and different forms of guava and papaya [58] to improve blood pressure, triglycerides, and markers of glycaemia, respectively. One intervention included an embargo policy, and targeted childhood malnutrition in communities [32]. 

### 3.2. Intervention Effectiveness

For this review, we classified effective studies as those having a statistically significant difference related to the outcomes of interest (*p* value <0.05) between intervention and control group or baseline and follow up. Overall, more than half (52%, *n* = 26) of the 50 included studies showed significant improvement on indicators of metabolic health or nutritional status, and a little less than a quarter (24%, *n* = 12) showed some improvement on indicators of metabolic health or nutritional status. Of these effective studies, 20 assessed metabolic outcomes and 6 assessed nutritional outcomes. The details of these studies are captured in Table 3.

#### 3.2.1. Effectiveness by Intervention Type

Effective metabolic-based outcomes (Table 3) included those with significant improvements in markers of glycemia [39,40,47,52,56,58,62,68], BMI [43,45,49,59,60], BP [24,42,61,67], weight change [54,57] and endothelial function [51]. Ten interventions were multifaceted, in other words, they included several modalities [39,45,52,57,59,60,61,62,67,68], five had an aspect of dietary change including direct changes to diet such as cooking method, calorie reduction, salt modification and change in dietary pattern (i.e., fasting) [24,40,43,47,56], two were based on nutrition education [49,54], two were based on specific foods [42,58], and one was based on a nutrition supplement [51]. Effective nutrition-based outcomes included those with significant improvements in markers of anaemia [25,31], weight for age and BMI for age z scores [38,71], vitamin D deficiency [50], and childhood stunting [28]. Four interventions were focused on nutritional supplements [25,28,31,50], one was multifaceted [71], and one was based on nutrition education [38]. 

Effective multifaceted interventions with metabolic outcomes incorporated an aspect of nutrition education [45,52,59,60,61,62,67,68], including study nutritionists to advise participants [39,57]. Most incorporated an aspect of physical activity such as educational sessions [45,59,60,61,68], promotion (i.e., by giving participants resistance bands) [52] or access to a physiotherapist [39,57], while some incorporated sessions on behaviour therapy or change [57,60,61,68], as well as stress reduction [45], medical support such as diabetic self-medication management sessions [39,62,68], and promotion of glucose monitoring through distribution of glucometer kits [52]. Two studies included practical aspects such as healthy cooking sessions [59] or a dietician-accompanied tour of a supermarket [60], while one included national campaigns to increase consumer awareness [59]. For the multifaceted intervention that assessed outcomes of nutritional status, nutrition education was combined with the provision of measured feeds [71].

#### 3.2.2. Effectiveness by Intervention Level and Setting

Of the twenty effective interventions with metabolic outcomes, thirteen were performed at an individual level, one at a household level [68], three at an institutional level [45,59,60] two at a community level [39,61] and only one at a national/policy level [67]. These targeted adults within a clinic [40,43,49,51,54,56,57,62,68], among the general population [24,39,47,58,67], at their workplace [59,60], and within a community setting [61]. One intervention targeted only women in a clinical setting [52], one had no setting information [42] and one targeted school children [45]. Similarly, most effective interventions with nutrition outcomes were conducted at the individual level [28,31,38,50,71], and one was conducted at the community level [25]. These targeted children at the community [25,28,31] or clinical setting [38,71], and adults among the general population [50].

#### 3.2.3. Effectiveness by Intensity

We adapted the definition of intervention intensity (frequency of intervention activities) developed by Nava et al., 2015, who categorised intervention intensity as high (greater than or equal to three intervention activities per week), moderate (bi-weekly to bi-monthly activities), and low (activities conducted no more than once monthly) [74]. The author explained that the level of participant exposure, determined by the intervention intensity, had a major influence on efficacy [74]. For this review, we categorised high intensity as short, medium, and long term, consisting of three or more activities weekly over three or less weeks, more than three to less than twelve weeks, and twelve or more weeks, respectively. Moderate and low intensity was sub-categorised as short or long term; moderate consisted of bi-weekly to bi-monthly activities over three or less months, and more than three months, respectively, and low intensity consisted of monthly activity over three months or less, and more than three months, respectively (Figure 3).

High intensity interventions that reported metabolic outcomes were more effective regardless of the study duration, described as long term [40,42,43,51,52,61,67], medium term [47], or short term [24,58]. Effective interventions with moderate intensity were primarily performed on a short term basis [45,59,60], while those of low intensity were either performed on a short term [39,42,54,56,57,58,59,60,61,62,63,64,65,66,67,68] or long term basis [62]. One intervention provided no information related to intensity. Notably, all effective interventions that reported nutrition outcomes had high intensity, either on a long term [25,28,50] or medium term basis [31,71], or moderate intensity on a long term basis [38].

### 3.3. Risk of Bias

Overall risk of bias for each record is described in Table 2, however all related domains for each tool used are reported in Appendix A [22,23]. Eleven studies (22%) had low risk, thirty-two (64%) had moderate risk or some concern, and seven (14%) had high or serious risk. The quality issues related to the six studies with high or serious risk [27,42,49,54,58,67] should be considered when applying the evidence, including designing future interventions. For the 26 effective studies, 2 had low risk [51,54], 14 had moderate risk [24,39,40,43,45,47,52,56,57,59,60,61,62,68] and 4 had high risk of bias [42,49,58,67] (metabolic outcomes); 4 had low risk [25,31,50,71], and 2 had moderate risk [28,31] (nutrition outcomes). 

## 4. Discussion

This systematic review identified 50 studies of nutrition-based interventions in SIDS aimed at improving nutritional status (*n* = 14) and metabolic health (*n* = 36). Although interventions varied by type, study aims, outcomes assessed, and quality, common features of the implemented interventions were those with a multifaceted design (42%; *n* = 21), conducted at the individual level (62%; *n* = 31), and targeted adults within a clinical setting (28%; *n* = 14). Overall, a total of 26 interventions were found to be effective. 

Other interventions were either ineffective or had mixed effects (combination of significant positive effects plus no change/insignificant effect on outcomes). Sixteen assessed metabolic outcomes and eight assessed nutritional outcomes. For those assessing metabolic outcomes, multifaceted interventions were most common, and authors attributed ineffectiveness of interventions to low-intensity of interventions, lack of nutrition education, and socio-cultural factors such as hierarchical structure and the role of gender, and highlighted that greater integration of strategies to address these, such as inclusion of families or group-based interventions as well as measures taken to build capacity within communities, may have resulted in more effective outcomes [35,36,44,55,59,69,70,73]. The development of low-cost interventions that may allow high-intensity intervention activities were also highlighted [35,36]. For interventions that assessed nutritional outcomes, supplement or fortified food-based interventions were most common. Similar to the interventions that assessed metabolic outcomes, these studies highlighted the importance of considering higher intensity interventions with ‘lower-cost’ supplement substitutes as well as environmental factors, to the perceived success of interventions within the setting of SIDS [27,29,30,37,41]. 

Twenty of the effective interventions assessed metabolic health outcomes, half of which had a multifaceted design that incorporated an aspect of nutrition education supplemented by a physical activity component including educational sessions and/or sessions on behaviour therapy, stress reduction, or medical support. Some interventions also incorporated practical aspects such as healthy cooking sessions. This was consistent with the literature conducted in other settings, whereby interventions incorporating nutrition education and the promotion of physical activity and/or behavior therapy with practical aspects such as recipe creation, and healthy cooking classes were effective on metabolic outcomes including weight change, BMI, blood glucose, and BP [75,76,77,78,79,80]. One study highlighted that a multifaceted lifestyle intervention was even more effective than metformin [80]. The promotion of these interventions was emphasised by a recent systematic review which stated that multifaceted lifestyle interventions encompassing aspects of diet as well as exercise are recommended as the first-line treatment for metabolic heath issues since they have positive long term-effects [81]. It is important to understand the underlying reasons for the interventions’ success, however, most studies give very little or no explanation. A multifaceted intervention that aimed to support diabetes self-management highlighted that most interventions led by community health workers utilised group-based models which often resulted in low attendance, since persons within underserved and low-resource settings may not be able to afford external group visits [68]. The authors attributed the success of their intervention to the integrated delivery which included at-home visits. They also attributed the intervention’s success to the culturally appropriate and linguistic adaption (simple terms and instructions) of tools and materials used, which was also highlighted in other effective interventions [24,39,47,49,52,58,61]. One study emphasised that culturally adapted interventions are effective because they use more accessible local resources and decrease the need for the involvement of highly trained professionals [39]. The need for culturally adapted interventions and efforts to negate out of pocket costs for participants is common to SIDS, since these settings are limited due to scarce resources. 

The other six effective interventions assessed nutritional outcomes, more than half (*n* = 4) of which used supplements aimed at improving indicators of ‘under-nutrition’. These included the use of micronutrient sprinkles and powder to improve anaemia, and lipid supplementation as well as Vitamin D supplements to improve growth and micronutrient deficiencies among children. This was also consistent with the literature conducted in other settings [82,83], which found that supplement-based interventions, specifically ready-to-use supplementary food, were effective in improving markers of anaemia, and wasting among children under 5 years of age [82,83]. Similar to the interventions discussed above, most authors highlighted that scarce resources were a major limitation within these settings [28,31,32,38,71]. One of the included studies specifically highlighted the success of an intervention that used micronutrient powder distributed by microfinance institutions [25]. The study highlighted that over 60% of child deaths could be prevented with access to health interventions and micronutrient supplements. However, in many SIDS, particularly those that are low or middle-income and/or with high rates of childhood stunting and wasting, these are inaccessible. The authors added that there was a need for the integration of microfinance institutions and healthcare, since it provides ease in reaching poor rural beneficiaries on a national scale, comprises pre-established supply chains and linkages funded by the institutions, and can facilitate mass delivery of micronutrient powders or other health-related products, which decreases total cost drastically. Another effective intervention which focused on reducing severe malnutrition in children spoke of the importance of utilising pre-existing resources. A major aspect of the intervention was related to supervised feeding of children, and although the authors highlighted that improving the nursing: patient ratios were difficult, this issue was addressed by utilising the services of nursing students, when available [71].

Studies within SIDS and other settings also highlighted the positive long-term impact of these supplement-based interventions on nutritional status; however, it is important to note that within these studies, most follow up periods were less than six months post-intervention, and there was a lack of information on impact over longer periods [25,28,31,50,83]. In addition, we found that although most of the included supplement-based interventions (eight of eleven overall, and six of the seven effective interventions) that assessed nutritional status in SIDS settings targeted children, but many of these interventions targeted the elderly in other settings [84,85,86]. Study authors emphasised that beyond the malnutrition concern among children, malnutrition remains pervasive among older adults [85]. 

### 4.1. Feasibility of Population-Level Interventions in SIDS

The findings suggest that most effective interventions were those of high intensity (frequent intervention activities performed three or more times weekly). This was the case for interventions that assessed both metabolic and nutritional outcomes, with high intensity long term interventions (over 12 weeks) being more prevalent. This was similarly found in other studies that highlighted the efficacy of high intensity interventions to improve both metabolic health and nutritional status [87,88]. The primary benefit of being enrolled in high intensity interventions was the maintenance of improved outcomes, such as significant weight loss over follow up points of two to four years post-intervention [87,88]. Notably, some of the high intensity effective interventions required high input of human resources and tended to be expensive, which may not be feasible to implement on a larger scale within the SIDS setting. 

However, the double burden of under- and over-nutrition related diseases faced by SIDS, discussed in the FAO plan of action in 2014 and subsequent GAP in 2017, highlights the need for policies and interventions to address these issues on a large scale or population level [5]. Our findings revealed that most of the interventions targeting both children and adults (*n* = 31; 62%) including the effective ones (thirteen of twenty that assessed metabolic health outcomes; six of six that assessed nutritional status outcomes) were conducted on the individual level. The feasibility of implementing interventions on a larger scale or population level was discussed in most studies, where cost was highlighted as a limiting factor. One author noted that the successful implementation of interventions aimed at treating diabetes was challenging as the cost of pharmaceuticals was prohibitive. For interventions that were presumed costly or where cost was not assessed by the authors, a recommendation was made that follow-up studies should be conducted to assess cost-effectiveness [28,52,59,60,67]. Iannotti et al., who assessed the effectiveness of a lipid-based nutrient supplement to improve linear growth (stunting) in young children, highlighted that the need for the participation of the mother in the intervention might have incurred real costs in relation to wages lost [28]. Another study that assessed the effectiveness of a team weight-loss challenge highlighted that direct costs per participant was estimated at USD 311 per person, and that it was worthwhile examining whether a program of this nature could be conducted on a national level if cost would decrease with mass participation (59). However, authors of low-budget effective interventions highlighted that the implementation of these would be feasible at a population level or larger scale [25,31,45,49,54,68]. This includes interventions that targeted aspects of over-nutrition such as obesity and diabetes by utilising nutrition education/education related to lifestyle changes taught by community leaders, nurses, or in-hospital staff [45,49,68], as well as nutrition or diabetic self-management guidance via phone-calls [54]. Interventions that targeted under-nutrition (malnutrition) that were deemed feasible on a population level included those that utilised low-cost micronutrient sprinkles and powders [25,31]. An effective intervention that utilised micronutrient sprinkles to decrease anaemia in children highlighted that the total cost of a two-month supply per child was USD 2 [31]. In addition, the only effective population level intervention, which addressed issues related to over-nutrition, highlighted the need to address complex issues with complex solutions. The study aimed to reduce salt intake among Fijians, and utilised a multi-level approach where food manufacturers, retailers, media, community leaders, nutrition staff and healthcare workers were engaged [67]. 

Studies conducted in other settings supported interventions conducted on a large scale, within the community-setting or on the population level that did not require substantial commitment and resources [89]. It is highlighted that these interventions also had the positive impact of social support [90,91,92]. Some studies found that interventions with a component of social support facilitated important changes in health-related behaviors among adults [93], while successful nutrition interventions targeting children were peer-based or family-based including setting family-based goals, modifying food environment at home or school, as well as hands-on approaches to teaching nutrition through games and group-based activities [94,95].

### 4.2. Interventions Addressing Both ‘under and over’ Nutrition

Of the 26 effective interventions, 1 aimed to address both malnutrition and overweight, by utilising a causal model that focused on factors affecting child development, survival, and growth, developed by the United Nations Children’s Fund (UNICEF). The intervention was described as an intermediary one since it was education-based and aimed to modify both mother and childcare practices. It intercepted at the point of antenatal care among groups of pregnant women, that met bi-monthly over five months to discuss health and nutrition during pregnancy. After childbirth, bi-monthly home visits were carried out to support breastfeeding and general childcare. In addition, group meetings and home-visits continued once per month to discuss child health such as vaccination, micronutrient supplements, growth monitoring and complementary feeding. Given that the size of the study was greater than 400 participants, the authors surmised that the results were positive and that a larger study was warranted. They also highlighted that the effectiveness of the intervention revealed the potential of simple interventions aimed at preventing under-nutrition and obesity, conducted by lay volunteers, or community nurses [38]. 

## 5. Limitations

This review was conducted utilising a transparent approach under PRISMA guidelines, however, highlighted by the findings of the ROB assessment, the included studies are subject to a quality limitation, due to their implementation in low-resource settings. It is important to note that there are differences within the heath care systems among SIDS (some being more developed, i.e., Singapore), hence interventions conducted within these settings may not be generalisable to other SIDS. Furthermore, the heterogeneity of included studies, small sample size of several interventions, and non-randomised design of some interventions that were included, also impacts generalizability, affects translation of the evidence, and may have led to selection bias. Overall, there was paucity of data, specifically a lack of interventions that assessed nutritional outcomes. We also found that there was a lack of evidence on population-wide interventions (baring two studies that focused on salt reduction in Fiji and an embargo in Haiti); for example, there was no evidence of the impact of sugar-sweetened beverages (SSB) taxes on nutrition or metabolic health. 

## 6. Conclusions

To inform future interventions and policies in efforts to address the double burden of diseases related to over- and under-nutrition faced by SIDS, it is important to assess the evidence of interventions within this setting and understand the setting’s limitations. This review provides a summary of the evidence of the impact of nutrition-based interventions aimed at improving metabolic health and nutrition status. Although the quality of included studies and assessed outcomes varied, the findings suggest that several approaches have been taken to address issues separately; however, multifaceted approaches including nutrition education at the early stages, targeting parent and child, may be effective to both issues. However, it also suggests that there is a lack of evidence for the impacts of population-wide and policy type measures on nutrition and/or metabolic health. In addition, the literature highlights a need for culture-appropriate interventions utilising community leaders and other non-exhaustive local means, due to resource scarcity faced by SIDS. Since nutritional disease burdens faced by SIDS affect a large proportion of the populations, efforts must be made to promote population level interventions. Further, efforts to translate evidence into polices or practice must acknowledge variation of issues unique to each country.

## Figures and Tables

**Figure 1 nutrients-14-03529-f001:**
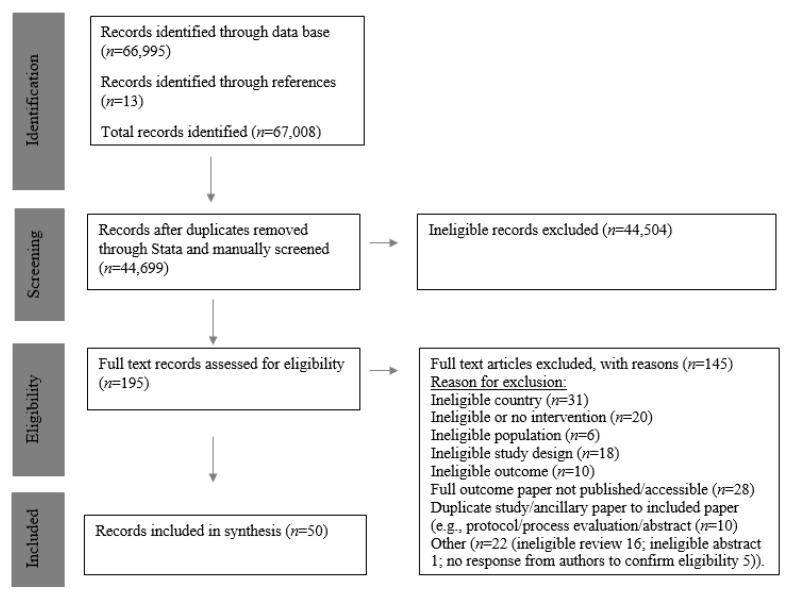
Preferred reporting items for systematic reviews and meta-analyses (PRISMA) flow chart.

**Figure 2 nutrients-14-03529-f002:**
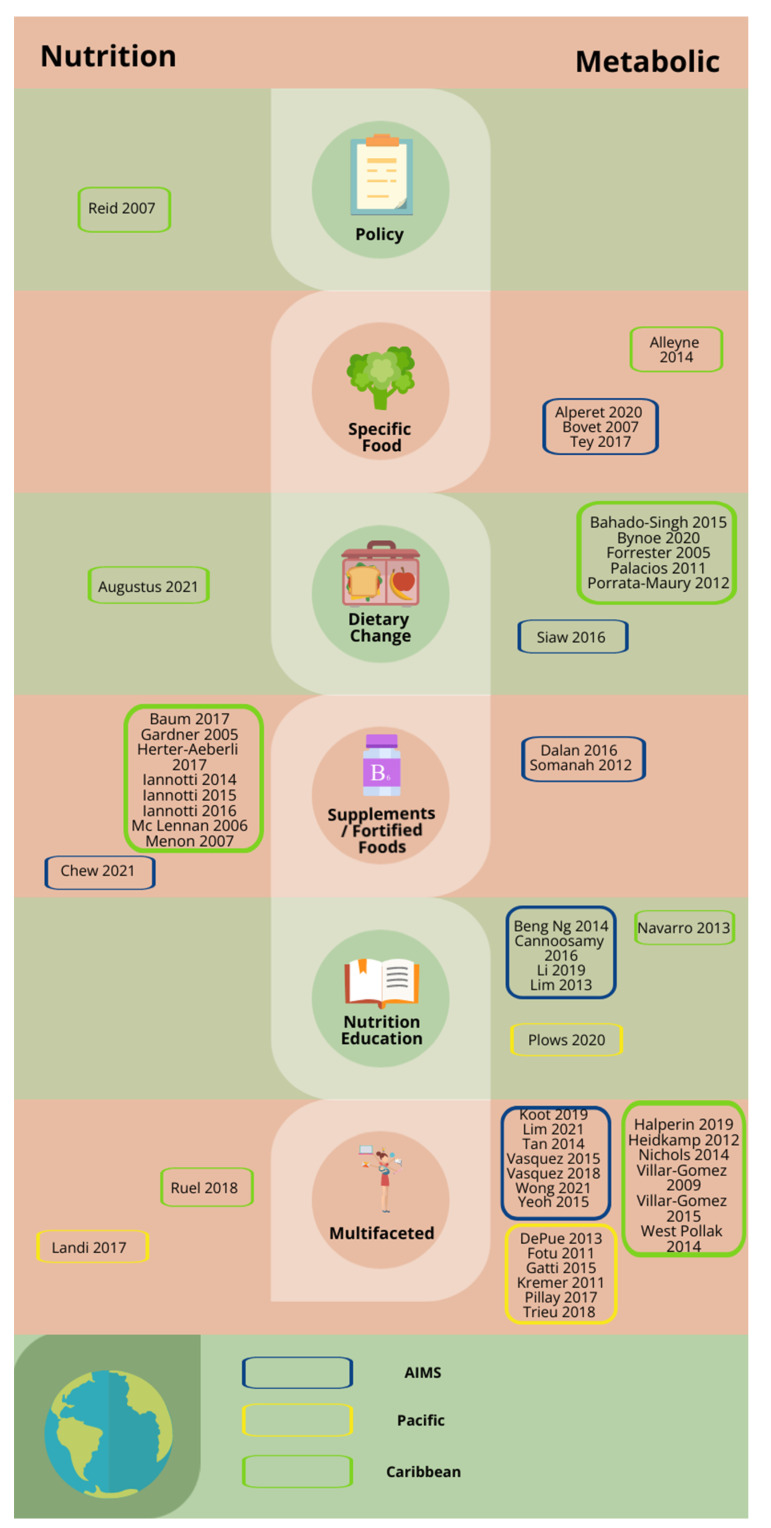
Nutrition-based interventions in SIDS by study location, intervention type and main outcome assessed (nutrition vs. metabolic). Studies were separated by geographic location and colour coded: AIMS; Pacific; Caribbean.

**Figure 3 nutrients-14-03529-f003:**
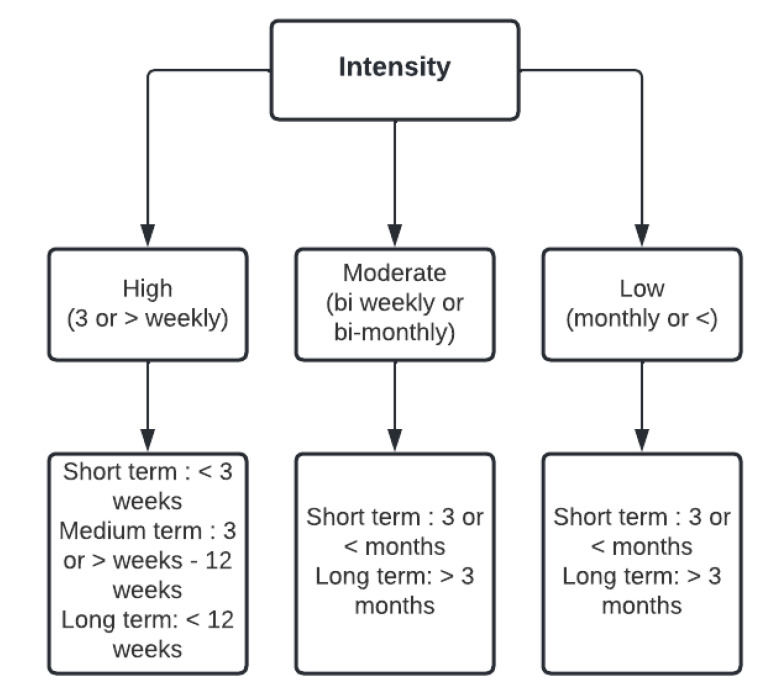
Categorisation of intervention intensity (activity frequency).

**Table 1 nutrients-14-03529-t001:** Eligibility criteria.

Criteria	Include	Exclude
**Study design**	Any experimental, quasi, or natural experimental designs (i.e., randomised controlled trials, non-randomised trials)	Non-intervention studies (i.e., cross-sectional studies, reviews, case reports, editorials)
**Outcome**	Quantitative outcomes—impact on any aspect of nutrition status or metabolic health	Dietary intake or nutrition knowledge as only outcome
**Study setting**	58 UN SIDS [19]	Any country that is not a SIDS
**Publication status**	Published and unpublished literature	Not applicable
**Language**	No language barriers: search terms were in English	Not applicable
**Time**	Articles from 1 January 2000 to 1 August 2020	All articles before and after the dates specified
**Participant characteristics**	Any ethnicity, age, gender, SIDS location, and socio-economic status.	Not applicable

**Table 3 nutrients-14-03529-t003:** Overview of effective interventions by intervention type (*n* = 26).

Ref.	Intervention Description	Setting	Intervention Level	Participant Intensity	Main Outcome Assessed
Specific food					
[42]	Impact of cocoa on blood pressure	No info	Individual	Low	Metabolic—blood pressure
[58]	Effect of consuming different forms (bite size, puree) and two fruit types (guava, papaya) on glycaemic response	General population	Individual	High	Metabolic—fasting blood glucose
Supplements/Fortified Foods					
[25]	Impact of micronutrient powders on haemoglobin concentration	Community	Community	High	Nutrition—haemoglobin concentration
[50]	Impact of nutritional supplement on vitamin D	Combination (general public, community centers, senior activity centers, polyclinics, hospitals, and by referrals from healthcare professionals in Singapore)	Individual	High	Nutrition—vitamin D deficiency
[51]	The effect of cholecalciferol supplementation on endothelial function	Clinic	Individual	High	Metabolic—endothelial function
[28]	Lipid supplement impact on childhood stunting	Community	Individual	High	Nutrition—stunting
[31]	Effectiveness of micronutrient sprinkles on anaemia	Community	Individual	High	Nutrition—anaemia
Nutrition Education					
[49]	To pilot test therapeutic lifestyle counselling on weight reduction	Clinic	Individual	Low	Metabolic—BMI
[54]	To evaluate nutritional outcomes after using an ANS service	Clinic	Individual	Low	Metabolic—weight Change
[38]	Lifestyle counselling to improve childhood malnutrition and overweight	Community	Individual	Moderate	Nutrition—weight change for age/height z scores
Multifaceted Intervention					
[68]	Impact of multifaceted intervention to support diabetic self-care	Clinic	Household	Low	Metabolic—HbA1C
[45]	To assess impact of multimodality weight gain prevention intervention	School	Institutional	Moderate	Metabolic—BMI
[52]	To assess the potential effectiveness and feasibility of Glycoleap (food coaching app)	Clinic	Individual	High	Metabolic—HbA1C
[67]	Intervention aimed at reducing population level salt intake.	General population	National/policy	High	Metabolic—blood pressure
[57]	Impact of non-surgical weight loss program	Clinic	Individual	Low	Metabolic—weight change
[59]	To assess the effectiveness of a national corporate team-based weight loss intervention	Workplace	Institutional	Moderate	Metabolic—BMI
[60]	To promote individuals to lose weight in a healthy way through a multi-component intervention.	Workplace	Institutional	Moderate	Metabolic—BMI
[39]	Lifestyle management of prediabetes and diabetes	General population	Community	Low	Metabolic—HbA1C
[61]	To improve the health behaviours and outcomes among women aged 50 years and older	Community	Community	High	Metabolic—blood pressure
[62]	To study the changes in body composition and metabolic profile in Muslim patients	Clinic	Individual	Low	Metabolic—HbA1C
Dietary Change					
[43]	Impact of ketogenic diet on nutritional status	Clinic	Individual	High	Metabolic—BMI
[40]	Impact of traditional Caribbean foods, with pronounced differences in GI on metabolic parameters	Clinic	Individual	High	Metabolic—HbA1C
[47]	Eight-week liquid (760 calorie) diet.	Combination (persons with type 2 diabetes recruited through a combination of publicity and contacts with local government and private healthcare providers)	Individual	High	Metabolic—fasting blood glucose
[24]	Impact of low-salt or high-salt diet on BP	General population	Individual	High	Metabolic—blood pressure
[56]	Impact of fasting and other nutrition related intervention on blood glucose	Clinic	Individual	NA	Metabolic—HbA1C

Table is stratified by intervention type (specific food, supplements/fortified food, nutrition education, multifaceted, and dietary change).

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
