# Peer review of "The Impact of Nutrition-Based Interventions on Nutritional Status and Metabolic Health in Small Island Developing States: A Systematic Review and Narrative Synthesis"

_nutrients, 2022, doi:10.3390/nu14173529_

Round 1
Reviewer 1 Report
To evaluate the impact of nutrition-based interventions on nutritional status and metabolic health in Small Island Developing States, this author conducted a systematic review of 50 eligible interventions. And this author found, twenty-six studies were effective. I thought this manuscript deal with interesting topic. However, I also found there are many inappropriate study design and description in present manuscript. I commented as following.
In abstract
1. According to title of present study, this is a systematic review. However, no concrete result which was calculated by statistical analysis were shown.
2. Since this author focus on the twenty-six studies out of 50 eligible interventions when describing the main part of their opinion, their conclusion could be strongly influenced by selection bias. This means present description did not image the results from systematic review.
3. Intervention type (suitable for the issues related to “over nutrition” or “under nutrition”) stratified analysis might be necessary. And those concrete conclusions should be shown in abstract.
4. Many abbreviations were used without spelling out.
5. Present abstract did not describe about the newly informative knowledge.
6. As the first paragraph of this abstract, nutrient poor and energy dense diets is most important topic in small island developing states. Therefore, this author should conduct systematic review with investigated nutrient poor and energy dense diets separately.
Introduction section
7. As described in the last paragraph of introduction section, recent systematic scoping review concluded that approaches utilized of their setting were inconsistent. Then this author should describe the superiority of present systematic review than that of previous study. The number and results were different from previous study never support those superiority. If intervention is the only superiority point, the potential reason why previous study showed incontinent results should be described in this section.
8. As described in introduction section, this author tried to identify best practice and approaches by systematically reviewing the interventional study. However, there is no description what this author made to achieve those intentions.
In method section
9. This author described as “Due of the heterogeneity of studies, related to intervention type, study design, participants, outcomes and measures, a meta-analysis to estimate a pooled effect size was not possible. Therefore, a narrative or descriptive analysis.” Therefore, this manuscript is not a systematic review but a narrative review.
Discussion
10. Since this manuscript is not systematic review, this author should describe as a narrative review or descriptive review manner.
Author Response
"Please see the attachment."

Reviewer 2 Report
The manuscript entitled ‘’The impact of nutrition-based interventions on nutritional status and metabolic health in Small Island Developing States: a systematic review.’’ is a systematic review about the effectiveness of nutrition-based interventions on nutritional status and metabolic health among per-sons in Small Island Developing States. The manuscript respect all criteria for this type of paper (PRISMA guidelines). It is well written, well structured, even it is a little bit long article. It is the result of the civilization consequences over the public health. Such article is important for health policies.
The studies included in the systematic review are cited in a special manner in references. I have no other comments. The article deserve to be published.
Author Response
"Please see the attachment."

Round 2
Reviewer 1 Report
Thanks to this author’s great effort, this revised manuscript became well improved. I have no more comment.